# WavJourney: Compositional Audio Creation with Large Language Models

## Abstract

Despite breakthroughs in audio generation models, their capabilities are often confined to domain-specific conditions such as speech transcriptions and audio captions. However, real-world audio creation aims to generate harmonious audio containing various elements such as speech, music, and sound effects with controllable conditions, which is challenging to address using existing audio generation systems. We present WavJourney, a novel framework that leverages Large Language Models (LLMs) to connect various audio models for audio creation. WavJourney allows users to create storytelling audio content with diverse audio elements simply from textual descriptions. Specifically, given a text instruction, WavJourney first prompts LLMs to generate an audio script that serves as a structured semantic representation of audio elements. The audio script is then converted into a computer program, where each line of the program calls a task-specific audio generation model or computational operation function. The computer program is then executed to obtain a compositional and interpretable solution for audio creation. Experimental results suggest that WavJourney is capable of synthesizing realistic audio aligned with textually-described semantic, spatial and temporal conditions, achieving state-of-the-art results on text-to-audio generation benchmarks. Additionally, we introduce a new multi-genre story benchmark. Subjective evaluations demonstrate the potential of WavJourney in crafting engaging storytelling audio content from text. We further demonstrate that WavJourney can facilitate human-machine co-creation in multi-round dialogues. To foster future research, the code and synthesized audio are available at: `https://anonymous.4open.science/w/WavJourney_Anonymous/`.

## 1 Introduction

The emerging field of multi-modal artificial intelligence (AI), a realm where visual, auditory, and textual data converge, opens up fascinating possibilities in our day-to-day life, ranging from personalized entertainment to advanced accessibility features. As a powerful intermediary, natural language shows great potential to enhance understanding and facilitate communication across multiple sensory domains. Large Language Models (LLMs), which are designed to understand and interact with human language, have demonstrated remarkable capabilities in acting as agents (Shen et al., 2023; Gupta & Kembhavi, 2023), engaging with a broad range of AI models to address various multi-modal challenges. While LLMs are regarded as effective multi-modal task solvers, an open question remains: can these models also become creators of engaging, realistic multimedia content?

Multimedia content creation involves digital media production in multiple forms, such as text, images, and audio. As a vital element of multimedia, audio not only provides context and conveys emotions but also fosters immersive experiences, guiding auditory perception and engagement. In this work, we address a novel problem of *compositional audio creation with language instructions*, which aims to automatically generate audio storytelling content using various elements such as speech, music, and sound effects from textual descriptions. Prior works have leveraged generative models to synthesize audio context that aligns with task-specific conditions, such as speech transcriptions (Panayotov et al., 2015), music descriptions (Agostinelli et al., 2023), and audio captions (Kim et al., 2019). However, the capabilities of these models to generate audio beyond such conditions are often limited, falling short of the demand for audio creation in real-world scenarios. In

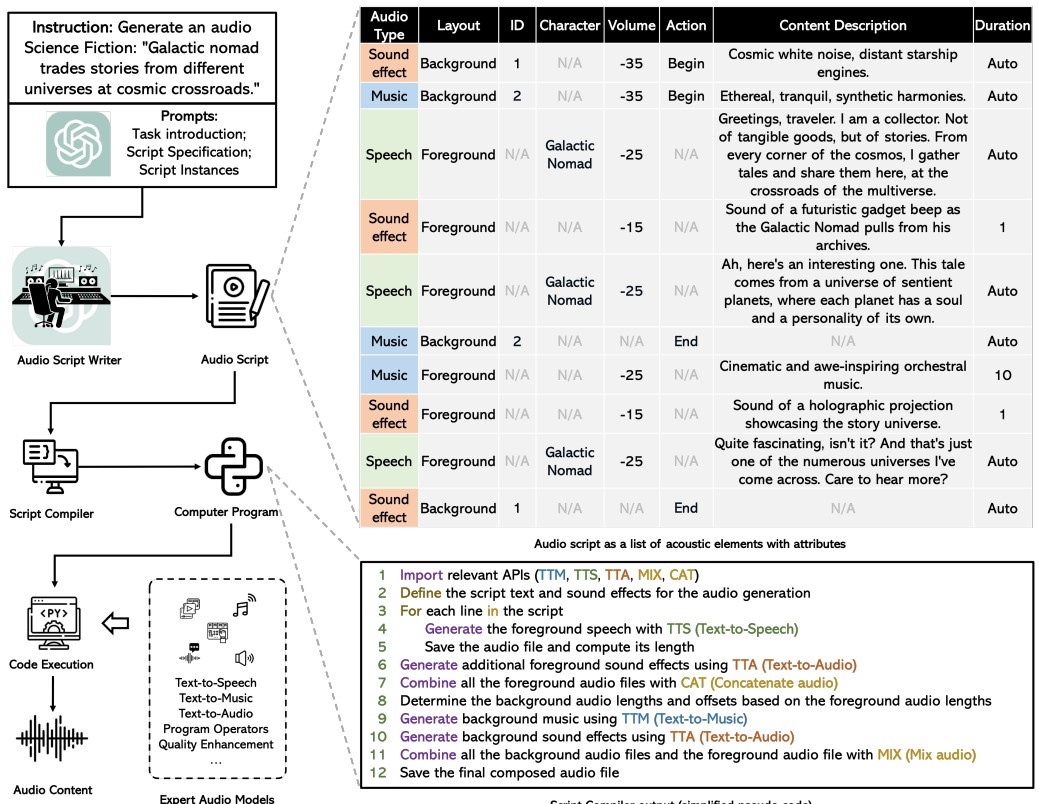

Figure 1: The overview of the WavJourney. The LLM is first prompted to be an audio script writer. As a conceptual representation of audio, the audio script provides the user with an interactive and interpretable interface. The audio script is then compiled using a script compiler and executed as a computer program. The execution process is powered by a set of expert audio generation models. The example illustrates a Sci-Fi audio storytelling creation given the text instruction: *"Generate an audio Science Fiction: Galactic nomad trades stories from different universes at cosmic crossroads."*

light of the integrative and collaborative capabilities of LLMs, it is intuitive to ask: can we leverage the potential of LLMs and these expert audio generation models for compositional audio creation?

Compositional audio creation presents inherent challenges due to the complexities of synthesizing intricate and dynamic audio content. Harnessing LLMs for compositional audio creation presents a range of challenges: 1) *Contextual comprehension and design*: using LLMs for compositional audio creation requires them not only to comprehend textual instructions but also to design audio story-lines featuring speech, music, and sound effects. How to expand the capabilities of LLMs in text generation to audio storytelling is a challenge; 2) *Audio production and composition*: unlike vision and language data, an audio signal is characterized by dynamic spatio-temporal relationships among its constituent audio elements. Leveraging LLMs for integrating various audio generation models to produce sound elements and further compose them into a harmonious whole presents additional challenges; 3) *Interactive and interpretable creation*: Establishing an interpretable pipeline to facilitate human engagement is critical in automated audio production, as it enhances creative control and adaptability, fostering human-machine collaboration. However, designing such an interactive and interpretable creation pipeline with LLMs remains an ongoing question.

We introduce WavJourney, a novel framework that leverages LLMs for compositional audio creation guided by language instructions. Specifically, WavJourney first prompts LLMs to generate a structured audio script, which includes the content descriptions of speech, music, and sound effects while considering their spatio-temporal relationship. To handle intricate auditory scenes, WavJourney decomposes them into individual acoustic elements along with their acoustic layout. The audio script is then fed into a designed script compiler, converting it into a computer program. Each line of

program code invokes a task-specific audio generation model, audio I/O functions, or computational operation functions. The computer program is subsequently executed to generate and compose the audio content. The overview of WavJourney with an example illustrates a science fiction audio storytelling creation is shown in Figure 1.

WavJourney offers multiple benefits for audio creation: as it is 1) *Contextually adaptive*, by leveraging the understanding capability and generalizable knowledge of LLMs, WavJourney can design audio storytelling scripts with diverse sound elements, and intricate acoustic relationships; 2) *Compositional*, benefiting from its composable design, WavJourney can automatically decompose complex auditory scenes into independent sound elements, enabling the integration of various task-specific audio generation models to controllably create audio content. Our approach differs from previous methods (Liu et al., 2023a; Huang et al., 2023a), where end-to-end generation often fails to generate audio aligned with complex textual descriptions; 3) *Training-free*, as WavJourney eliminates the need for training audio generation models or fine-tuning LLMs, making it resource-efficient; 4) *Interactive*, The interpretability offered by both the audio script and computer program facilitates audio producers with varying expertise to engage with WavJourney, fostering human-machine co-creation in real-world audio production. WavJourney advances audio creation beyond traditional task-specific conditions and opens up new avenues for computational creativity in audio production.

Our contributions are summarized as follows:

- We present WavJourney, a framework that leverages LLMs for compositional audio creation. Given textual instructions, WavJourney can create audio storytelling content encompassing speech, music, and sound effects, without the need for additional training.
- We assess WavJourney's capability on the AudioCaps (Kim et al., 2019) and Clotho (Drossos et al., 2020) text-to-audio generation benchmarks. Results show that WavJourney outperforms previous state-of-the-art methods in both subjective and objective evaluations. Notably, WavJourney's synthesized audio even outperforms the ground truth audio of AudioCaps in subjective tests, suggesting a potential capability to produce realistic and contextually relevant audio from texts.
- We present a multi-genre storytelling benchmark along with various subjective metrics such as engagement, creativity and emotional resonance. Subjective evaluations show that WavJourney can craft audio storytelling with positive scores in subjective metrics and outperforms state-of-the-art text-to-audio synthesis models.
- We further demonstrate that WavJourney enables interactive audio creation in multi-round dialogues, facilitating human-machine co-creation in audio production applications.

## 2 RELATED WORK

**Large Language Models (LLMs)**. LLMs such as GPT-3 (Brown et al., 2020), LLaMA (Touvron et al., 2023a), and ChatGPT (OpenAI, 2022) have advanced the research area of natural language processing (NLP) due to their capability to generate human-like text. Recently, LLMs have emerged as agents, demonstrating their capability to address intricate AI tasks by integrating a range of domain-specific AI models. ViperGPT (Suris et al., 2023) and VisProg (Gupta & Kembhavi, 2023) have demonstrated the significant promise of LLMs in decomposing complex vision-language tasks such as visual reasoning and text-to-image generation. These methods can generate a computer program (e.g., Python code) for each decomposed sub-task, which is executed sequentially to offer an explainable task solution. HuggingGPT (Shen et al., 2023) leverages ChatGPT (OpenAI, 2022) as a controller to manage existing AI models in HuggingFace (HuggingFace, 2016) for solving AI tasks in the domain of language, vision, and speech. Similar to HuggingGPT, AudioGPT (Huang et al., 2023b) connects multiple audio foundation models to solve tasks with speech, music, sound understanding, and generation in multi-round dialogues. In the context of existing research, considerable efforts have been dedicated to leveraging the integrative and collaborative capabilities of LLMs to solve multi-modal tasks. However, there remains a relatively unexplored area concerning the potential of LLMs in audio content creation.

**Audio Creation**. The process of audio creation is complex and dynamic, involving various components such as content design, music composition, audio engineering, and audio synthesis. Traditional methods have relied on human-involved approaches such as field recording (Gallagher, 2015),

Foley art (Wright, 2014), and music composition (Tokui et al., 2000) co-existing with digital signal processing modules (Reiss, 2011). In recent years, the intersection of AI and audio creation has gained significant attention. AI-driven approaches, particularly generative models, have demonstrated promising results in synthesizing audio content for speech (Tan et al., 2022; Wang et al., 2017; Wu et al., 2023), music (Agostinelli et al., 2023; Copet et al., 2023), sound effects (Liu et al., 2023a; Huang et al., 2023a; Liu et al., 2021; Yuan et al., 2023b), or specific types of sounds, such as footsteps or violin (Bresin et al., 2010; Engel et al., 2020). Existing audio generation models primarily focus on synthesizing audio content based on a particular type of task condition, such as speech transcriptions (Panayotov et al., 2015), music descriptions (Agostinelli et al., 2023), or audio captions (Kim et al., 2019), and they are not designed to generate compositional audio containing speech, music, and sound effects with controllable textual conditions. Leveraging data-driven approaches for addressing compositional audio creation is resource-intensive. It demands the collection of a sophisticated audio dataset with corresponding text annotations, as well as the training of powerful audio models.

## 3 WAVJOURNEY

WavJourney is a collaborative system composed of an audio script writer utilizing LLMs, a script compiler and a set of audio generation models such as zero-shot text-to-speech[1], text-to-music, and text-to-audio generation models. The overall architecture is illustrated in Figure 1. The pipeline of WavJourney can be deconstructed into two major steps: 1) *Audio script generation*: given a text instruction, the audio script writer initiates the process by warping the input instruction with specific prompts. Then, the LLM is engaged with these prompts, which directs it to generate an audio script conforming to the structured format. 2) *Script compiling and program execution*: Subsequently, the script compiler transcribes the audio scripts into a computer program. The computer program is further executed by calling the APIs of expert audio generation models to create audio content. We describe the details of these two steps in the following sections.

### 3.1 AUDIO SCRIPT GENERATION

The first challenge in harnessing LLMs for audio content creation lies in generating an audio narrative script based on the input text instructions that often only contain conceptual and abstract descriptions. Recognizing that LLMs have internalized generalizable text knowledge, we utilize LLMs to expand input text instructions into audio scripts, including detailed descriptions of decomposed acoustic contexts such as speech, music, and sound effects. To handle spatio-temporal acoustic relationships, we prompt LLMs to output the audio script in a structured format composed of a list of JSON nodes. Each JSON node symbolizes an audio element, including acoustic attributes (e.g., duration and volume). In this way, a complex auditory scene can be decomposed into a series of single acoustic components. Then, we can create the desired audio content by leveraging diverse domain-specific audio generation models. We introduce these details in the following paragraphs.

**Format of Audio Script.** We define three types of audio elements: speech, music, and sound effects. For each audio element, there are two types of layouts: foreground and background. Foreground audio components are concatenated sequentially, i.e. no overlap with each other. Background audio on the other hand can only be played along with some foreground elements, i.e. they can not be played independently (overlaps of background audio are allowed). Sound effects and music can either be foreground or background, while speech can only be foreground. The concepts above are reflected in the format of a list consisting of a series of JSON nodes, with each node embodying a unique audio component. Each node is supplemented with a textual description of its content. To enhance the auditory experience, each audio component is assigned a volume attribute. Sound effects and musical elements are furnished with an additional attribute pertaining to length, facilitating control over their duration. For speech components, a character attribute is assigned. This attribute enables the synthesis of personalized voices in the later stage, thereby enhancing the narrative thread of audio storytelling. This layered structuring and detailing of audio elements contribute to creating rich and dynamic audio content. We observe that the outlined format is able to cover a wide variety

---

[1]Zero-shot text-to-speech (TTS) refers to the ability of a TTS system to generate speech in a voice that has not been explicitly trained on, given an unseen voice preset as a condition for zero-shot synthesis.

of auditory scenarios and its simplified list-like structure facilitates the understanding of LLMs for complex auditory scenes. An example of the audio script is shown in the Listing 1.

**Personalized Voice Setting.** Advances in zero-shot TTS have facilitated the synthesis of personalized speech based on specific voice presets. In WavJourney, we leverage zero-shot TTS to amplify the narrative depth of audio storytelling, utilizing a set of voice presets tailored for diverse scenarios. More specifically, each voice preset is provided with descriptions of its characteristics and appropriate application scenarios. In subsequent stages, we can leverage LLM to allocate a suitable voice from the preset to each character outlined in the audio script. We utilize a simple prompt design to facilitate the voice allocation process, as discussed in the next paragraph. This design enables WavJourney to create personalized auditory experiences.

**Prompt Strategy.** To generate rich and formatted audio scripts from given text instructions, we wrap the text instructions within a prompt template. The prompt template contains the specifications of each JSON node type, including audio type, layout, and attributes. The prompt template is shown in Table 5 in Appendix A.2. The instructions listed in the prompt template facilitate formatting the audio script generated by LLMs. In the next step, following the receipt of the generated audio scripts, we instruct the LLM to parse the characters outlined in each speech node into personalized speech presets. The prompt used for voice parsing is shown in Table 6 in Appendix A.2. The audio script with parsed voice mapping is further processed through the script compiler, generating the code for subsequent stages.

## 3.2 Script Compiling and Program Execution

WavJourney employs a script compiler to automatically transcribe the audio script into a computer program. The pseudo-code of the script compiler is described in Appendix A.4. Each line of compiled code in the program invokes a task-specific audio generation model, audio I/O function, or computational operation function (e.g., mix, concatenate). The program is subsequently executed, resulting in an explainable solution for compositional audio creation. In contrast to previous studies that utilize LLMs to generate code (Suris et al., 2023; Gupta & Kembhavi, 2023), WavJourney prompts LLMs to generate textual audio scripts, which foster improved comprehension for audio producers without programming expertise. Additionally, the process of crafting a computer program to compose intricate auditory scenes requires an elaborate series of procedures within the code to manage audio length calculations and the mixing or concatenation operations of audio clips. Given the unpredictability of LLMs (Shen et al., 2023), there are occasions when they may fail to adhere to specified instructions during the generation process. By introducing a script compiler, we can mitigate the potential exceptions in the program workflow arising from the instability of LLMs, thereby reducing this uncertainty during inference.

## 4 Audio Storytelling Benchmark

We introduce a multi-genre story benchmark with five storytelling genres in real-world contexts: *education*, *radio play*, *romantic drama*, *science fiction (Sci-Fi)*, and *travel exploration*. For each genre, we prompted ChatGPT (OpenAI, 2022) to generate ten story titles, each ranging from 15 to 25 words. This diverse set of generated stories is designed to be a benchmark for evaluating WavJourney's capabilities in audio storytelling creation. The story examples can be found in Table 1. Inspired by expert-driven metrics design process (Shah et al., 2003), we further design a subjective evaluation protocol for comprehensively assessing the generated audio storytelling using five metrics: *engaging*, *creativity*, *relevance*, *emotional resonance*, and *pace & tempo*. Each metric is scored from 1 to 5, and the details of each metric are described in Table 2. Our subjective evaluation protocol is developed with input from audio and product experts, allows for an assessment that goes beyond traditional evaluations (e.g., Mean Opinion Score (Ribeiro et al., 2011)) to consider narrative complexity, music and sound design, and vocal delivery in a coherent and comprehensive manner.

| Genre | Story Title Example |
|---|---|
| Education | *"Mathematics in Nature: Exploring Fibonacci Sequences and Golden Ratios"* |
| Radio Play | *"Ella and Sean, in playful debate, as pastries crumble and cappuccinos steam"* |
| Romantic Drama | *"Secrets whispered, emotions swell, two hearts navigating love's turbulent sea"* |
| Sci-Fi | *"Mars colonists find ancient alien artifacts; Earth's history is not ours alone"* |
| Travel Exploration | *"Iceland's geysers and elves: a land where nature's fury meets mythical tales"* |

Table 1: Examples of multi-genre audio storytelling benchmark.

| Score | Engaging | Creativity | Relevance | Emotional Resonance | Pace & Tempo |
|---|---|---|---|---|---|
| 1 | Not at all | Not at all | Not at all | Not at all | Too Slow |
| 2 | Slightly | Slightly | Slightly | Slightly | Slightly Slow |
| 3 | Moderately | Moderately | Moderately | Moderately | Just Right |
| 4 | Very | Very | Very | Very | Slightly Fast |
| 5 | Extremely | Extremely | Extremely | Extremely | Too Fast |

Table 2: Subjective evaluation protocol for audio storytelling creation.

## 5 EXPERIMENTS

### 5.1 EXPERIMENTAL SETUP

**WavJourney Setup.** We utilize the GPT-4 model (OpenAI, 2022) as the LLMs for WavJourney. For text-to-music and text-to-audio generation, we adopt the publicly available state-of-the-art models MusicGen (Copet et al., 2023) and AudioGen (Kreuk et al., 2022), respectively. As for text-to-speech synthesis, we leverage the Bark (Suno, 2023) model, which can generate realistic speech and is able to match the tone, pitch, emotion, and prosody of a given voice preset. We use four voice presets[2] drawn from Bark's official voice presets[3] as the WavJourney's default voice settings. To enhance the quality of synthesized speech, we apply the speech restoration model VoiceFixer (Liu et al., 2022) after the Bark model. We use 16 kHz sampling rate for processing audio signals. We implement the computer program in the Python language. For the volume control of the generated audio content, we adopt the Loudness Unit Full Scale (LUFS) standard (Steinmetz & Reiss, 2021).

**Text-to-Audio Generation Evaluation.** We assess the performance of WavJourney in text-to-audio generation on AudioCaps (Kim et al., 2019) and Clotho (Drossos et al., 2020) benchmarks. For comparison, we use two publicly available state-of-the-art text-to-audio generation models: AudioGen (Kreuk et al., 2022) and AudioLDM (Liu et al., 2023a) as the baseline systems. All audio samples were sampled at 16 kHz for evaluation. Datasets and baseline reproduction details are described in the Appendix A.1. We use both objective and subjective methods for evaluation. To control the length of WavJourney-generated audio for fair comparison, we simply add the duration condition as a suffix for input prompts (e.g., "the duration of generated audio must be 10 seconds." ).

For objective evaluation, in line with previous works (Kreuk et al., 2022; Liu et al., 2023a; Yuan et al., 2023a), we adopt the objective metrics: Frechet Audio Distance (FAD), Kullback-Leibler Divergence (KL), Inception Score (IS) for evaluation. FAD calculates the Frechet distance between the distribution of the embedding value of two audio groups, extracted through a pre-trained VGGish model (Hershey et al., 2017). KL presents the similarity between the logit distributions of two groups of audio calculated by an audio tagging model, Patch-out Transformer (Koutini et al., 2021). IS illustrates the variety and diversity of the target audio group. A higher IS indicates a larger variety with vast distinction, while both KL and FAD indicate better audio fidelity with lower scores. Objective metrics were computed for each model across the AudioCaps and Clotho test sets.

For subjective evaluation, following previous work (Kreuk et al., 2022), we adopt metrics: Overall Impression (OVL) and Audio and Text Relation (REL). Both OVL and REL metrics have a Likert scale (Likert, 1932) between one and five, where a larger number indicates better performance. Furthermore, we perform a preference-based subjective evaluation where listeners compare audio

---

[2]Male voices: 'v2/en_speaker_1', 'v2/en_speaker_6'; Female voices: 'v2/en_speaker_9', 'v2/de_speaker_3'.
[3]https://github.com/suno-ai/bark/tree/main/bark/assets/prompts

files from two different systems using the same text description, choosing the one that sounds better considering its corresponding text condition. In the subjective evaluation, we used 50 audio files, randomly sampled from the AudioCaps and Clotho test sets, respectively. In all subjective tests, WavJourney is compared against the baseline systems as well as the ground truth (GT) audio.

**Audio Storytelling Creation Evaluation.** We evaluate the capability of WavJourney in audio storytelling creation on the story benchmark using the subjective evaluation protocol introduced in Section 4. For baselines, we compare WavJourney with AudioGen and AudioLDM systems. Each generated audio story lasts approximately 30 to 60 seconds.

## 5.2 RESULTS ON TEXT-TO-AUDIO GENERATION

**Results on AudioCaps**. WavJourney outperforms AudioGen and AudioLDM systems in all subjective tests on the AudioCaps benchmark. As shown in the left part of Table 3, WavJourney achieves an OVL score of 3.75 and a REL score of 3.74. Compared with AudioLDM and AudioGen, WavJourney has 0.36 and 0.19 higher in OVL, and 0.4 and 0.22 higher in REL, respectively. WavJourney even marginally surpasses the ground truth audio in the OVL (3.73) metric and is on par in the REL (3.76) metric. The subjective preference-based testing results are consistent with the OVL and REL results, as shown on the left side of Figure 2. Specifically, against AudioGen, WavJourney is superior in approximately 40% of the cases, while AudioGen outperforms WavJourney in 27.2% of instances. No distinguishable difference is noted in 34.7% of the cases. When compared to AudioLDM, WavJourney excels in roughly 48.7% of the cases, lags in about 24.4%, and both share comparable performance in 26.9% of scenarios. The comparison against ground truth indicates that WavJourney surpasses in 35.6% of cases, they both match without clear differentiation in 40%, and WavJourney is only inferior in 24.4% of the instances. Notably, WavJourney is the first audio generation system that can *exceed real-world audio performance on the AudioCaps benchmark*, suggesting its capability to generate realistic and contextually relevant audio content from texts. We found that WavJourney's objective evaluation results slightly underperform compared to AudioGen. This contrasts with our subjective assessments, suggesting that *the objective evaluation (e.g., FAD) may not always be an effective way to evaluate the performance of text-to-audio generation systems, which is consistent with conclusions drawn in previous work*. (Choi et al., 2023).

**Results on Clotho**. WavJourney surpasses AudioGen and AudioLDM systems across all three objective metrics on the Clotho benchmark, as shown in the right part of Table 3. Specifically, AudioGen demonstrates better performance over AudioLDM with scores of 2.55, 2.21, and 7.41 for FAD, KL, and IS metrics, respectively. WavJourney achieves a FAD score of 1.75, a KL score of 2.18, and an IS score of 9.15. The subjective evaluation results are consistent with the objective metrics. AudioGen and AudioLDM demonstrate comparable performance, achieving an OVL score of 3.41 and a REL score of approximately 3.37. WavJourney outperforms both by a significant margin, with an OVL score of 3.61 and a REL score of 3.56. The performance of WavJourney and ground truth is closely approximated, where the ground truth audio files have high quality sourced from the Freesound platform (Font et al., 2013). Specifically, the performance gap is merely 0.1 and 0.15 for OVL and REL metrics, respectively, demonstrating the strong performance of our proposed system. WavJourney achieves *a new state-of-the-art on the Clotho benchmark* in terms of both objective and subjective metrics.

The results of subjective preference-based testing are shown on the right side of Figure 2. WavJourney outperforms AudioGen in 42.8% of cases while being outperformed in 25.6%. Compared with AudioLDM, WavJourney leads in 44.5% of cases but lags in 26.5%. There is no clear difference in roughly 30% of cases for both systems. The comparison conclusion against AudioGen and AudioLDM is coherent with the above objective and subjective evaluation results. However, the ground truth audio outperformed WavJourney in a substantial 63.7% of the cases, indicating there is potential space for improvement on the Clotho benchmark.

**Case Studies and Analysis**. We randomly select two audio clips with complex captions (e.g., more than 15 words, containing at least three sound events) from the AudioCaps test set. We further perform case studies to study the effectiveness of WavJourney, compared with the AudioLDM and AudioGen systems. The results and comparison are shown in Figure 3, and synthesized audio files are available on the project page (Anonymous, 2023). We manually localize each sound event in the spectrogram using the colored line underneath per spectrogram for visualization. AudioGen and

| Model | AudioCaps | | | | | Clotho | | | | |
|---|---|---|---|---|---|---|---|---|---|---|
| | FAD ↓ | KL ↓ | IS ↑ | OVL ↑ | REL ↑ | FAD ↓ | KL ↓ | IS ↑ | OVL ↑ | REL ↑ |
| AudioLDM | 4.65 | 1.89 | 7.91 | 3.39 | 3.34 | 3.57 | 2.19 | 6.84 | 3.41 | 3.36 |
| AudioGen | **2.15** | **1.49** | **8.15** | 3.56 | 3.52 | 2.55 | 2.21 | 7.41 | 3.41 | 3.37 |
| WavJourney | 3.38 | 1.53 | 7.94 | **3.75** | **3.74** | **1.75** | **2.18** | **9.15** | **3.61** | **3.56** |
| Ground Truth | - | - | - | 3.73 | 3.76 | - | - | - | 3.71 | 3.71 |

Table 3: Performance comparison on the AudioCaps and Clotho benchmarks. The best values from AudioLDM, AudioGen, and WavJourney systems are shown in bold, and when a value exceeds Ground Truth, it will be underlined to highlight it.

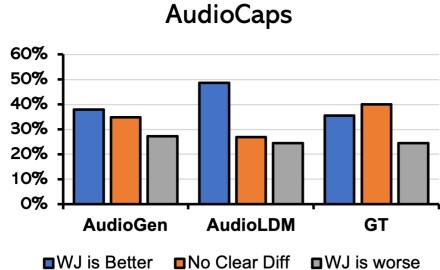
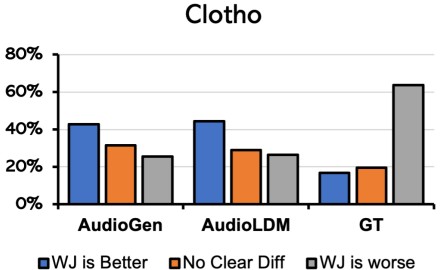

Figure 2: Preference-based subjective evaluation on AudioCaps and Clotho benchmarks. WJ is the abbreviation of WavJourney.

AudioLDM fail to generate audio that aligns with textual descriptions and often results in unintelligible synthesized speech. WavJourney not only successfully generates all desired audio elements, including intelligible speech, but can also organize each audio element in the proper temporal-spatio layout as described in the text (e.g., two events happening simultaneously when instructed by the word "as" or one sound is behind another when the word "follow" appears). The compositional design offers WavJourney better subjective results compared with AudioLDM and AudioGen systems, which also suggests the capability of WavJourney for controllable audio generation with textually-described semantic, temporal, and spatial conditions.

## 5.3 RESULTS ON AUDIO STORYTELLING CREATION

Subjective evaluation results are shown in Table 4. WavJourney performs well in generating audio storytelling with appropriate pacing and tempo and achieves positive scores i.e., marginally above the moderate level for aspects including engagement, creativity, relevance, and emotional resonance for all genres. Compared with baselines, WavJourney consistently outperforms both AudioLDM and AudioGen across all evaluated aspects. The subjective evaluation results demonstrate the practicality of WavJourney in real-world audio production applications and indicate its potential in crafting engaging audio storytelling from texts. We will release all the synthesized audio files for future comparison. Synthesized audio clips are available on the project page (Anonymous, 2023).

| Genre | Engaging | Creativity | Relevance | Emotional Resonance | Pace & Tempo |
|---|---|---|---|---|---|
| AudioGen | 2.09 | 2.17 | 2.18 | 2.09 | 2.56 |
| AudioLDM | 2.12 | 2.18 | 2.18 | 2.23 | 2.50 |
| WavJourney | **3.28** | **3.18** | **3.52** | **3.04** | **2.97** |

Table 4: Subjective evaluation results on the proposed multi-genre audio storytelling benchmark.

## 5.4 HUMAN-MACHINE CO-CREATION

The natural interface of the audio script is designed to enable audio producers to actively engage in automated audio creation. By leveraging the communication capabilities of LLMs, WavJourney

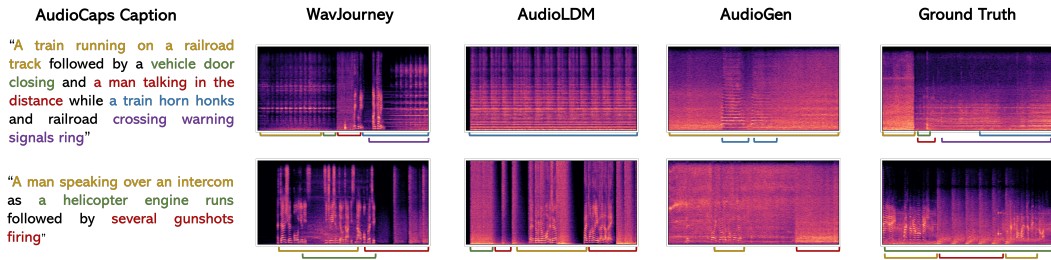

Figure 3: Case studies on AudioCaps benchmark.

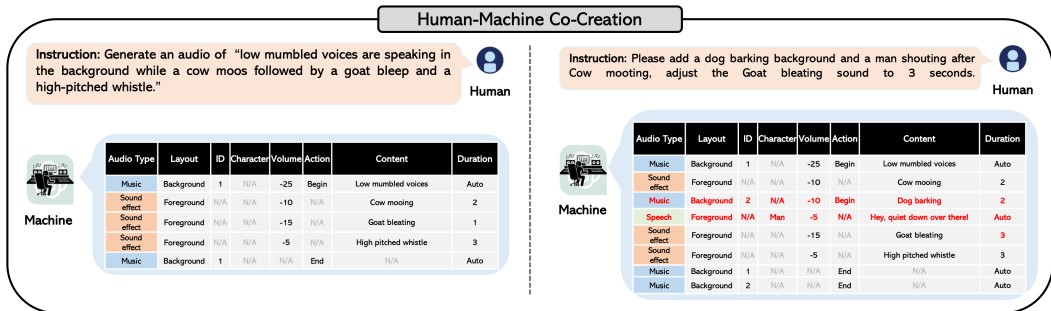

Figure 4: An illustrative example of human-machine co-creation.

allows users to customize the audio content through multi-round dialogues. We perform several case studies on the AudioCaps dataset. As shown in Figure 4. Initially, WavJourney was requested to synthesize the audio content conditioned on the provided text descriptions, followed by dynamic changes to the audio elements through multi-round dialogues, such as adding new audio elements (e.g., "add a dog barking background and a man shouting after cow mooting") or modifying acoustic attributes (e.g., "adjust the goat bleating sound to 3 seconds"). Furthermore, we provide another case study on radio play storytelling creation, with the focus on speech content control, as shown in Figure 5 and 6. The successful execution of these tasks by WavJourney demonstrates its great promise in fostering human-machine co-creation.

## 6 LIMITATIONS

Although WavJourney can create audio content with text instructions, limitations could be observed as follows: 1) *Extensibility*: WavJourney relies on structured audio scripts to represent auditory scenes and script compilers to generate computer programs, which is inflexible to expand its functional capabilities; 2) *Artificial composition*: The process of decomposing and re-mixing audio may result in synthetic audio that deviates from real-world sound distributions, particularly concerning music composition, which requires the alignment of beats, chord progression, and melody in multiple music tracks; 3) *Efficiency*: The reliance of WavJourney on LLMs and multiple audio generation models introduces time costs when generating complex audio content. Improving the efficiency of WavJourney could facilitate its practicality for real-world applications.

## 7 CONCLUSION

In this work, we presented WavJourney, which connects LLMs with diverse expert audio generation models, enabling compositional audio creation via natural language instructions. WavJourney can schedule various expert audio generation models to create audio content by decomposing auditory scenes into individual audio elements with their acoustic relationships. Extensive experiments suggest that WavJourney is can generate realistic audio from captions and also demonstrate great potential in crafting engaging audio storytelling from the text. WavJourney opens up new avenues for advancing Artificial Intelligence Generated Content (AIGC) in the realm of audio content creation.

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

# A  APPENDIX

## A.1  DETAILS OF TEXT-TO-AUDIO GENERATION EVALUATION

**AudioCaps Dataset** AudioCaps is the largest publicly available audio captioning dataset contains 50 725 10-second audio clips sourced in AudioSet. As the AudioSet is sourced from YouTube, the quality of audio clips in the AudioCaps is diverse. AudioCaps is divided into three splits: training, validation, and testing sets. The audio clips are annotated by humans with natural language descriptions through the Amazon Mechanical Turk crowd-sourced platform. Each audio clip in the training sets has a single human-annotated caption, while each clip in the validation and test set has five ground-truth captions. We retrieved AudioCaps based on the AudioSet we downloaded. Our retrieved AudioCaps dataset contains 49 274/49 837 audio clips in the training set, 494/495 clips in the validation set, and 929/975 clips in the test set.

**Clotho Dataset** Clotho is an audio captioning dataset that contains high-quality sound clips obtained from the FreeSound platform[4]. Each audio clip in Clotho has been annotated via the Amazon Mechanical Turk crowd-sourced platform. Particular attention was paid to fostering diversity in the captions during the annotation process. In this work, we use Clotho v2 which was released for Task 6 of the DCASE 2021 Challenge[5]. Clotho v2 contains 3839, 1045, and 1045 audio clips for the development, validation, and test split respectively. Each audio clip is annotated with five captions. Audio clips are of 15 to 30 seconds in duration and captions are 8 to 20 words long.

**Baseline Results Reproduction** We use the official implementations of AudioGen[6], AudioLDM[7] to synthesize audio from textual descriptions drawn from AudioCaps and Clotho test sets. For the AudioCaps dataset, which exclusively comprises 10-second audio clips, we maintain consistency by setting all our results to this same duration. In contrast, the Clotho evaluation set contains audio clips of varying lengths. To ensure fairness, we randomly select audio lengths ranging from 15 to 30 seconds for Clotho, reflecting the diverse nature of this dataset. For each audio in the AudioCaps and Clotho test sets, we use its first textual description as the input condition. As a result, for each model, we produced 929 samples for AudioCaps and 1045 audio clips for Clotho, respectively.

**Subjective Evaluation Setup.** Our subjective evaluations were carried out using Amazon Mechanical Turk[8], a renowned crowd-sourcing platform. We provided raters with detailed instructions and illustrative examples to ensure a consistent evaluation process. To maintain the reliability of our results, we imposed specific criteria for the participating workers: a minimum average approval rate of 60% and a history of at least 50 approved tasks. Each audio sample was assessed by a minimum of 10 different raters. The final score for each system was calculated by averaging scores across all raters and audio samples. We ensured fair compensation for the participants, with payments aligning with or exceeding the US minimum wage. The human subjective evaluation conducted in this work has received a favorable opinion from the ethics committee at the authors' affiliated institution.

---

[4] https://freesound.org/
[5] https://dcase.community/challenge2021
[6] https://github.com/facebookresearch/audiocraft/
[7] https://github.com/haoheliu/AudioLDM
[8] https://requester.mturk.com/

## A.2 Prompt Templates

---

Prompt #1

---

I want you to act as an audio script writer. I'll give you an instruction which is a general idea and you will make it an audio script in List format containing a series of JSON nodes.

The script must follow the rules below:

Each line represents an audio JSON node. There are three types of audio: sound effects, music, and speech. For each audio, there are two types of layouts: foreground and background. Foreground audios are played sequentially, and background audios are sound effects or music which are played while the foreground audio is being played.

Sound effects can be foreground or background. For sound effects, you must provide its layout, volume (dB, LUFS standard), length (in seconds), and detailed description of the sound effect. Example: {"audio_type": "sound_effect", "layout": "foreground", "vol": -35, "len": 2, "desc": "Airport beeping sound"}

Music can be foreground or background. For music, you must provide its layout, volume (dB, LUFS standard), length (in seconds), and detailed description of the music. Example: {"audio_type": "music", "layout": "foreground", "vol": -35, "len": 10, "desc": "Uplifting newsroom music"}

Speech can only be foreground. For speech, you must provide the character, volume (dB, LUFS standard), and the character's line. You do not need to specify the length of the speech. Example: {"audio_type": "speech", "layout": "foreground", "character":""News Anchor", "vol": -15, "text": "Good evening, this is BBC News"}

For background sound effects, you must specify the id of the background sound effect, and you must specify the beginning and the end of a background sound effect in separate lines, hence you do not need to specify the length of the audio. Example: {"audio_type": "sound_effect", "layout": "background", "id":1, "action": "begin", "vol": -35, "desc": "Airport ambiance"} ... {"audio_type": "sound_effect", "layout": "background", "id":1, "action": "end"}

For background music, it's the same as background sound effects.

The output format must be a list of the root node containing all the audio JSON nodes.

---

Table 5: Prompt template used for generating audio scripts.

---

Prompt #2

---

Given an audio script in json format, for each character that appeared in the "character" attribute, you should map the character to a "voice type" according to their lines and the "voice types" features. Each character must be mapped to a different voice type, and each "voice type" must be from one of the following:

- Female1: a normal female adult voice, British accent
- Female2: a normal female adult voice, American accent
- Male1: a normal male adult voice, British accent
- Male2: a normal male adult voice, American accent

Output should be in the format of CSV, like:
```
[character 1], [voice type 1]
[character 2], [voice type 2]
...
```

---

Table 6: Prompt template used for voice parsing.

## A.3 EXAMPLE OF AUDIO SCRIPT

```
1  AudioScript = [
2      {"audio_type": "music", "layout": "background", "id":1, "
           action": "begin", "vol": -30, "desc": "Dramatic orchestral
            news theme"},
3      {"audio_type": "speech", "layout": "foreground", "character":
           "News Anchor", "vol": -15, "text": "Welcome to Mars News
           ..."},
4      {"audio_type": "music", "layout": "background", "id":1, "
           action": "end"},
5      {"audio_type": "sound_effect", "layout": "foreground", "vol":
           -35, "len": 1, "desc": "Transition swoosh"},
6      {"audio_type": "speech", "layout": "foreground", "character":
           "Reporter", "vol": -15, "text": "We're here at the ..."},
7      ...
8      {"audio_type": "speech", "layout": "foreground", "character":
           "News Anchor", "vol": -15, "text": "... Stay tuned to Mars
            News for the latest updates."},
9      {"audio_type": "music", "layout": "foreground", "vol": -30, "
           len": 5, "desc": "orchestral news outro music"}
10  ]
```

Listing 1: Example audio script in the list form.

## A.4 SCRIPT COMPILER

The script compiler works with a list of JSON nodes, where each node represents an audio element that is categorized either as foreground or background, along with their attributes (e.g., volume, duration). The pseudo code of the script compiler is described in the Algorithm 1.

The algorithm performs the following six steps:

**Initialization**: The algorithm starts by initializing two lists, one for foreground audio nodes (foregroundAudioList) and another for background audio nodes (backgroundAudioList). It also initializes a variable to keep track of the next foreground node ID.

**Node Classification**: The algorithm iterates through the root list and checks each node's type. If the node is of type 'foreground', it is added to the foregroundAudioList, and the next foreground node ID is incremented. If the node is marked as the beginning of a background audio segment, it is assigned the next foreground node ID and added to the backgroundAudioList.

**Foreground Audio Processing**: The compiler then creates a list to store the lengths of the foreground audio segments. For each foreground audio node, the corresponding audio is generated and its length calculated and stored in the list.

**Background Audio Processing**: The algorithm calculates the target length for the background audio based on the foreground audio's start and end IDs, and then generates and calculates the length of the background audio.

**Audio Composition**: The algorithm combines all the foreground audio segments to create the final foreground audio. Then, for each background audio node, it mixes the background audio with the foreground audio, offsetting the background audio to sync with the foreground audio segments.

**Final Output**: The resulting final audio, which is a mix of foreground and background audio tracks, is outputted from the script compiler.

---

**Algorithm 1:** Pseudo code of Script Compiler

---

**Initialize:** foregroundAudioList as an empty list, backgroundAudioList as an empty list, nextForegroundNodeID as 0

1 **forall** *node in the root list* **do**
2      **if** *node.type is "foreground"* **then**
3          Add node to foregroundAudioList
4          Increment nextForegroundNodeID by 1
5      **else**
6          **if** *node.isBeginning is True* **then**
7              Set node.beginForegroundID as nextForegroundNodeID
8              Add node to backgroundAudioList
9          **else**
10              Set backgroundNode in backgroundAudioList with id node.id's endForegroundID as nextForegroundNodeID
11          **end**
12      **end**
13 **end**
**Initialize:** fgAudioLengths as an empty list
14 **forall** *foregroundAudio in foregroundAudioList* **do**
15      Generate the audio based on foregroundAudio
16      Calculate the generated length, append it to fgAudioLengths
17 **end**
18 Concatenate all generated foreground audio to create finalForegroundAudio
19 **forall** *backgroundAudio in backgroundAudioList* **do**
20      Calculate targetLength by using backgroundAudio's beginForegroundID, endForegroundID and fgAudioLengths
21      Generate the audio of targetLength based on backgroundAudio
22      Calculate the offset from the beginning of the finalForegroundAudio using beginForegroundID and fgAudioLengths, set as offset
23 **end**
**Initialize:** finalAudio as finalForegroundAudio
24 **forall** *backgroundAudio in backgroundAudioList* **do**
25      Mix finalAudio with backgroundAudio generated audio, with offset according to backgroundAudio.offset
26 **end**
27 Output finalAudio

---

## A.5 ABLATION STUDIES ON TEXT-TO-AUDIO GENERATION

**Motivation.** We conducted an ablation study on the text-to-audio generation task using the Audio-Caps and Clotho datasets. Our motivation is to investigate the sources of performance improvements, particularly those arising from 1) *the introduction of a text-to-speech synthesis model*, and 2) *the use of LLMs to decompose complex audio captions*.

**Experiments setup.** We manually checked the text captions of 50 audio clips from the AudioCaps and Clotho datasets, which were previously used in a subjective listening test (as detailed in Section 5.2). These clips were then classified into two categories: *Speech-Inclusive*, which includes speech and and other sounds, and *Non-Speech Exclusive*, which consists of non-speech elements only. In the case of AudioCaps, 23 (46%) audio clips were categorized as speech-inclusive and 27 (54%) as non-speech exclusive. For Clotho, the speech-inclusive and non-speech exclusive categories contained 26 (52%) and 24 (48%) audio clips, respectively. We decomposed the subjective evaluation results reported in table 3 into speech-inclusive and non-speech exclusive splits, detailed in Tables 7 and B 8 for AudioCaps and Clotho dataset, respectively.

**Results and anslysis.** Across both the AudioCaps and Clotho datasets, WavJourney consistently outperforms AudioLDM and AudioGen in terms of the OVL and REL metrics for both speech-inclusive and non-Speech exclusive splits. Particularly, given that WavJourney uses AudioGen for text-to-audio synthesis, the improvements in performance over the AudioGen system in the non-speech exclusive splits shows the benefits of WavJourney's use of LLMs to decompose complex audio captions. Moreover, WavJourney achieved impressive results in the speech-inclusive splits. Specifically, for AudioCaps, WavJourney obtained an OVL score of 3.93 and a REL score of 3.90, significantly surpassing both AudioLDM and AudioGen. These scores even exceed the Ground Truth scores of 3.77 and 3.85 for OVL and REL, respectively. For Clotho, WavJourney achieved an OVL score of 3.61, which substantially outperforms AudioLDM and AudioGen and is on par with the Ground Truth score of 3.60. We attribute this additional improvement to the introduction of text-to-speech synthesis model that can produce better speech component than AudioLDM and AudioGen text-to-audio generation systems.

An interesting question is raised, *why does WavJourney perform better than or on par with Ground Truth audio on speech-inclusive data*? We suggests that the improvements may come from three aspects: 1) The use of state-of-the-art text-to-speech synthesis models that can produce realistic and clear speech content. 2) Some audio clips in AudioCaps or Clotho datasets are of poor quality. As these datasets were originally designed for audio captioning research, their audio clips may not always yield high subjective evaluation results (e.g., audio clips in AudioCaps sourced from noisy YouTube videos). 3) The effective use of the contextual understanding capabilities of LLMs. When generating complex audio that includes speech, WavJourney can design the content of the speech related to the context[9], potentially leading to a better impression during the listening test, considering that in real-world recordings, the speech content may not always be related to the acoustic scene.

| Model | Speech-Inclusive (46%) | | Non-Speech Exclusive (54%) | | Overall | |
|---|---|---|---|---|---|---|
| | OVL ↑ | REL ↑ | OVL ↑ | REL ↑ | OVL ↑ | REL ↑ |
| AudioLDM | 3.25 | 3.21 | 3.50 | 3.45 | 3.39 | 3.34 |
| AudioGen | 3.53 | 3.57 | 3.57 | 3.47 | 3.56 | 3.52 |
| WavJourney | **3.93** | **3.90** | **3.59** | **3.60** | **3.75** | **3.74** |
| Ground Truth | 3.77 | 3.85 | 3.69 | 3.68 | 3.73 | 3.76 |

Table 7: Ablation study on AudioCaps speech-inclusive and non-speech exclusive splits.

---

[9]For example, in a scenario where the audio scene is a bustling city street, real-world recordings might include speech content unrelated to the scene, potentially causing a disconnect for the listener. However, WavJourney can generate complex audio with contextually relevant speech content for such a scene, such as a conversation about navigating the city or comments on city sounds. This context-aware approach enhances the listening experience and leads to improved results in listening tests.

| Model | Speech-Inclusive (52%) | | Non-Speech Exclusive (48%) | | Overall | |
|---|---|---|---|---|---|---|
| | OVL ↑ | REL ↑ | OVL ↑ | REL ↑ | OVL ↑ | REL ↑ |
| AudioLDM | 3.26 | 3.22 | 3.57 | 3.50 | 3.41 | 3.36 |
| AudioGen | 3.30 | 3.34 | 3.52 | 3.40 | 3.41 | 3.37 |
| WavJourney | **3.61** | **3.55** | **3.61** | **3.56** | **3.61** | **3.56** |
| Ground Truth | 3.60 | 3.67 | 3.81 | 3.75 | 3.71 | 3.71 |

Table 8: Ablation study on Clotho speech-inclusive and non-speech exclusive splits.

## A.6 TURING TEST ON TEXT-TO-AUDIO GENERATION

We conducted Turing test (Pease & Colton, 2011) experiments on AudioCaps and Clotho text-to-audio generation benchmarks. Participants were presented with audio clips and tasked with categorizing them as real, ambiguous, or fake. To mitigate any potential bias that might skew listeners' perceptions, we did not provide textual descriptions of the audio clips in the Turing test experiments. The audio clips utilized in this study are identical to those used in the subjective listening tests detailed in Section 5.2. Our intuition is to assess whether WavJourney can generate audio that is indistinguishable from real-world audio in human perception.

The results on the AudioCaps and Clotho are described in Table 9 and 10, respectively. WavJourney outperforms AudioGen and AudioLDM across both AudioCaps and Clotho in Turing tests. Specifically, WavJourney achieved the highest perceived-as-real rates (53.8% for AudioCaps and 54.0% for Clotho) and the lowest rates of being perceived as fake and ambiguous. However, WavJourney did not succeed in passing the Turing Test by matching the perceived-as-real rate of the Ground Truth audio, which was 65.8% for AudioCaps and 62.2% for Clotho. The good results achieved by WavJourney show that it performs better in generating realistic audio clips than AudioLDM and AudioGen, although there is still a gap between synthetic and real audio as judged by human listeners. The discrepancy between the results of the Turing test and other subjective metrics (e.g., OVL, REL, preference-based test) on AudioCaps may be attributed to text-to-speech generated speech. Although this generated speech can positively influence subjective metrics (as discussed in Section A.5), it may be easily to distinguish due to human's sensitive perception of speech.

| Model | Perceived as Real | Ambiguous | Perceived as Fake |
|---|---|---|---|
| AudioGen | 53.0% | 15.6% | 31.4% |
| AudioLDM | 47.2% | 18.8% | 34.0% |
| WavJourney | 53.8% | 13.4% | 32.8% |
| Ground Truth | 65.8% | 15.6% | 18.6% |

Table 9: Turing test on AudioCaps benchmark.

| Model | Perceived as Real | Ambiguous | Perceived as Fake |
|---|---|---|---|
| AudioGen | 46.4% | 18.8% | 34.8% |
| AudioLDM | 46.8% | 17.2% | 36.0% |
| WavJourney | 54.0% | 14.8% | 31.2% |
| Ground Truth | 62.2% | 14.0% | 23.8% |

Table 10: Turing test on Clotho benchmark.

## A.7 SYSTEM-LEVEL ABLATION STUDIES

In this section, we perform system-level ablation studies by comparing against other design choices. Our experiments include 1) employing open-source large language models (LLMs) like Llama for audio script writing; and 2) using LLM (e.g., GPT-4) to generate Python code. These comparison aim to evaluate the effectiveness and efficiency of the module design in WavJourney.

### A.7.1 ON OPEN-SOURCE LLMS FOR SCRIPT WRITING

This section explores the script writing capabilities of Llama2-70B-Chat Touvron et al. (2023b), an open-source LLM, chosen for its accessibility and potential adaptability in diverse script writing scenarios. We use the 50 audio storytelling prompts introduced in Section 4 for testing. We prompted Llama2-70B-Chat to generate audio scripts from input text instructions, and then to parse voices based on the generated audio script and the system voice preset. The resulting audio script and voice mapping were fed into the script compiler, which validated their accuracy. We compared the *Compilation Success Rate (CSR)* of Llama2-70B-Chat with that of GPT-4 used in WavJourney. To ensure a fair comparison, we used the same prompts as those used in WavJourney.

| LLM Script Writer | Compilation Success Rate |
|---|---|
| Llama2-70B-Chat | 10% |
| Llama2-70B-Chat$_{processed}$ | 54% |
| GPT-4 | 94% |

Table 11: Compilation Success Rates for Llama2-70B-Chat and GPT-4 in audio script writing.

The results are presented in Table 11. The Llama2-70B-Chat script writer achieved only a poor CSR of 10%. We observe that this was caused by the hallucination (Rawte et al., 2023) of Llama2-70B-Chat, which often generated unexpected contexts (e.g., a summary of what it does at the end). We further manually processed the outputs of Llama2-70B-Chat by removing the hallucinated contexts. The result obtained after this processing is denoted as Llama2-70B-Chat$_{processed}$. After processing, the CSR improved to 54%. However, this still fell short of the performance achieved by GPT-4, which had a good CSR of 94%. The disparity in performance between Llama2-70B-Chat and GPT-4 highlights the challenges involved in audio script writing and suggests that an additional effort (e.g., instruction tuning (Liu et al., 2023b)) may be required to use open source LLMs for this task.

We demonstrate several errors made by Llama2-70B-Chat and GPT-4. For Llama2-70B-Chat, the most common failure case is the incorrect output format i.e., the integration of JSON syntax and natural language description. The model generated a detailed script but failed to maintain a consistent JSON format, as seen in the inclusion of comments like *'[Background sound effect: Soft office ambient noise, -35 dB, end]'* within the JSON structure. This presents a difficulty for Llama2-70B-Chat in adhering to a strict JSON format when trying to provide descriptive audio elements. In addition, several generated scripts have incomplete JSON structure, lacking essential attribute (e.g., *'character' attribute for speech nodes*). This suggests limitations in the Llama2-70B-Chat's understanding of structured data formats, leading to incomplete or incorrect outputs. For GPT-4, few failure cases involve using undefined audio types, such as *'child_laughter'*, which is not recognized within the pre-defined audio types. This indicates a potential issue with the model's understanding of the specific constraints and vocabularies required for this task.

### A.7.2 ON LLM-BASED SCRIPT COMPILER

Previous works (Suris et al., 2023; Gupta & Kembhavi, 2023) have used LLMs to generate code aimed at addressing complex vision-language tasks via compositional inference. In this experiment, we implement a LLM-based script compiler for comparison. We use the GPT-4 (OpenAI, 2022) model as the LLM-based compiler in this ablation study. Specifically, we experiment with 50 audio scripts generated by WavJourney from the storytelling benchmark. We design a specialized prompt template to enable a GPT4-based script compiler to parse code from an audio script through in-context learning. The simplified description of this prompt template is described in Table 13. To evaluate the reliability and efficiency, we computed the Execution Error Rate (EER) and the Average Compiling Time (ACT) for both the hand-crafted and GPT4-based compilers across a set of 50 examples. Experiments were carried out on a CPU machine equipped with an AMD EPYC 7502 32-Core Processor. The practical compiling time for the GPT4-based compiler is depend on the on-demand request and the length of audio script.

The results are presented in Table 12. The GPT4-based compiler has a significantly higher EER of 56%, along with an ACT of 63.16 seconds. This suggests that the GPT4-based compiler is less efficient and stable compared to the hand-crafted compiler, which demonstrated no execution errors

and an ACT of 0.03 seconds. The results show that our proposed hand-crafted compiler mitigates the instability in the script compilation and also greatly improve the inference efficiency.

| Script Compiler | Execution Error Rate | Average Compiling Time (seconds) |
|---|---|---|
| Hand-crafted | 0% | 0.03 |
| GPT4-based | 56% | 63.16 |

Table 12: Performance comparison of hand-crafted and GPT4-based script compilers in terms of Execution Error Rate (EER) and average Compiling Time (ACT).

| Simplified Prompt #3 |
|---|

I want you to act as an audio script compiler. I'll provide you with instructions for compiling an audio script in JSON node list format into a Python program.

[Audio Script Format Description]

[Description of the Python API Functions (TTM, TTS, TTA, MIX, CAT)]

[An Audio Script Example]

[A Compiled Python Code Example]

Instruction: compile the Python code given the audio script.

Table 13: Simplified description of the prompt template used for GPT4-based script compiler.

### A.8 The Analysis of Inference Cost

As discussed in Section 6, WavJourney uses LLM and multiple audio generation models, which introduces an additional time cost when generating complex audio content. To analyze the time cost, we randomly selected 20 text captions from the AudioCaps test set and calculated the average inference time required by WavJourney and AudioGen to generate 10-second audio clips. For WavJourney, we report the time costs associated with script writing and audio generation. The testing was conducted on a machine equipped with a single NVIDIA GeForce RTX 2080 Ti GPU. The results are shown in Table 14. In practice, by carefully optimizing the use of computational resources (e.g., parallel inference) for WavJourney, it is possible to achieve high-quality synthetic audio generation while minimizing the time required for inference, which we leave as future work.

| Model | Script Writing Time (s) | Audio Generation Time (s) |
|---|---|---|
| AudioGen | - | 23.0 |
| WavJourney | 8.1 | 45.3 |

Table 14: The comparison of inference cost for WavJourney and AudioGen systems.

## A.9 HUMAN-MACHINE CO-CREATION (SPEECH CONTENT CONTROL)

A case study on a radio-play-based storytelling example focusing on voice content control. In the example, we can control the content of the speech by topic or specific sentences and the type of language in the multi-turn conversation.

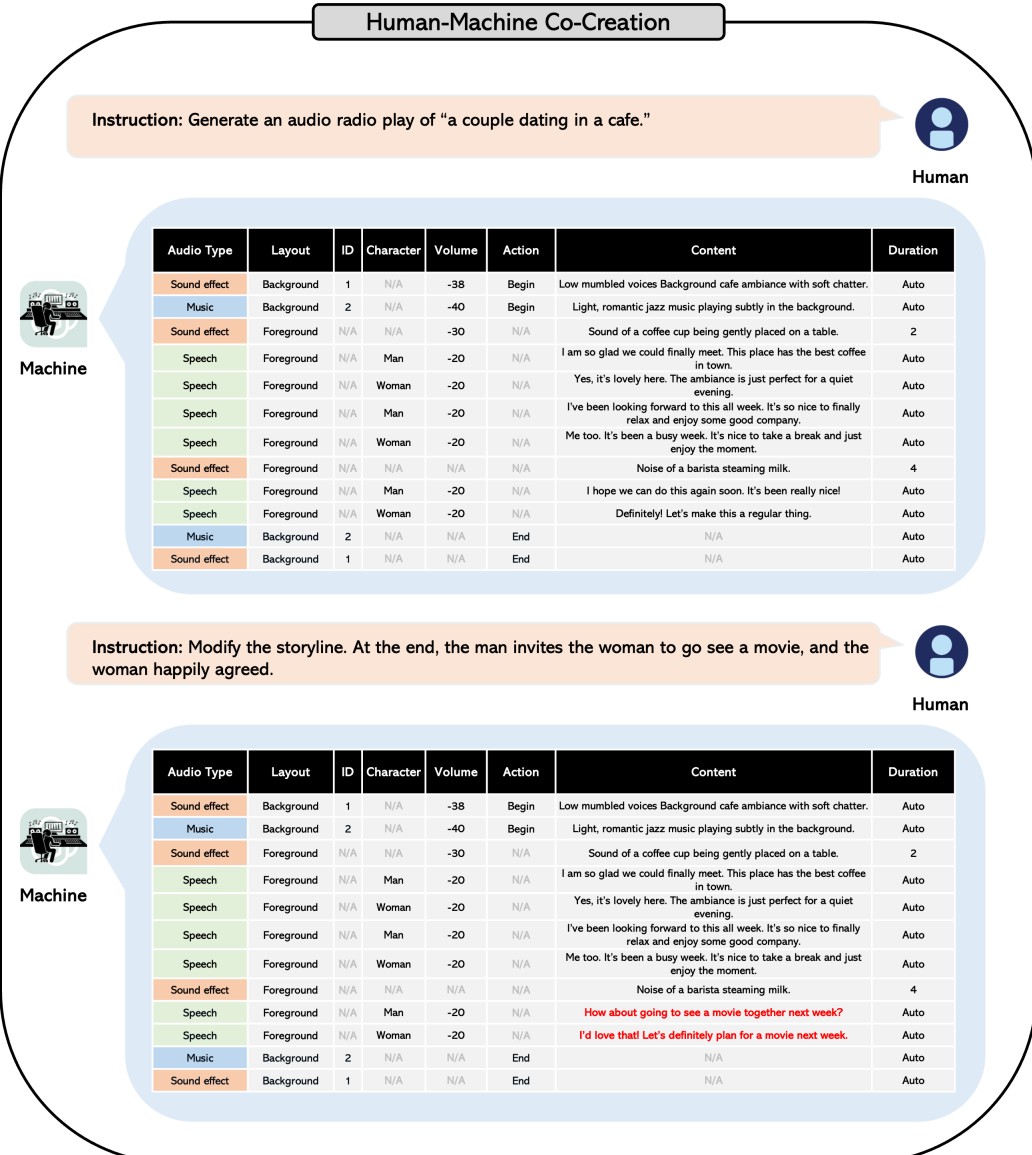

Figure 5: An example of human-machine co-creation with the focus on speech content control.

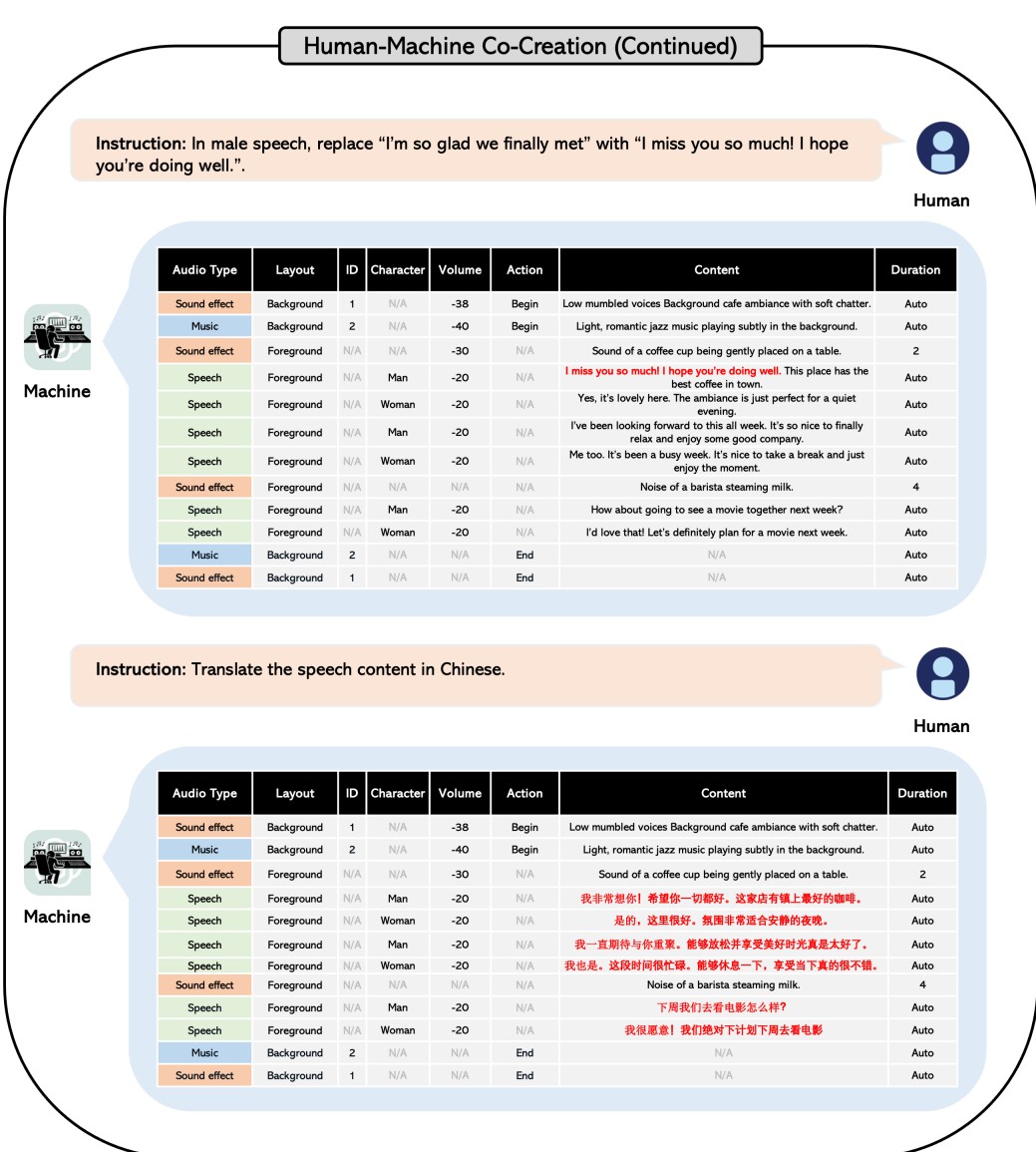

Figure 6: Continuation of Figure 5.

