# OpenReview forum: "WavJourney: Compositional Audio Creation with Large Language Models"
_ICLR.cc/2024/Conference — Submitted to ICLR 2024_

### Official Review · Reviewer_jF76 · 2023-10-26

**Soundness:** 2 fair
**Presentation:** 3 good
**Contribution:** 2 fair
**Rating:** 6
**Confidence:** 3

**Summary:**

The paper presents an interesting integrated system pipeline for producing storytelling audio by leveraging the power of LLM, i.e., GPT-4, along with different audio expert models.

At its core, the pipeline consists of an audio script writer, script compiler and a code executor to generate audio utilizing audio expert models. The input to the system is a short description (prompt) of an audio topic in natural language, and GPT-4 takes the prompt and converts it into event-based (speech, music and sound effects) audio script representation. Such script representation is then compiled by the manually designed script compiler, whose outputs are executables that could be invoked by different audio expert models.

Experimental results show that the method could outperform or achieve on-par results against several state-of-the-art text-to-audio models on AudioCaps and Clotho benchmarks. In addition, the paper presents a new type of benchmark  covering 5 genres, namely audio storytelling benchmark. Human evaluation on the method shows reasonable mean opinion scores in 5 different aspects.

**Strengths:**

1. The paper proposes a novel task, text-to-audio storytelling. The task opens up a huge space for AI-powered general audio creation.

2. The proposed system outperforms or achieves on-par  performance against previous SOTA methods, i.e., AudioGen and AudioLDM, in both objective and subjective metrics.

3. The new established benchmark is another good contribution.

4. The developed prototype and demos look amazing.

**Weaknesses:**

1. The method proposed has limited methodology contribution to the research community. The system reflects a crafted engineering pipeline by combining several off-the-shelf audio expert models through LLM. There is limited contribution in terms of machine learning algorithms.

2. As sort of an ensemble of expert models, the paper does not include any system-level ablation studies by comparing against other design choices.

    a. How much does the system rely on the capabilities of LLMs? How would it perform if LLM is not GPT-4, but rather those open-source alternatives like Llama.

    b. The script writer acts like an agent. There are numerous works and existing methods working on connecting LLM to perform various tasks, e.g., AutoGPT. But the authors didn’t compare against these agent-based system design.

**Questions:**

See weakness.

---

> ### Author Response · Authors · 2023-11-23
> **Response to Reviewer jF76 [system-level ablation studies ]**
>
> We would first like to thank the reviewer for the insightful review. Please find our responses to the questions in terms of system-level ablation studies below.
>
> >”How much does the system rely on the capabilities of LLMs? How would it perform if LLM is not GPT-4, but rather those open-source alternatives like Llama.”
>
> We evaluated the system using both GPT-4 and Llama2-70B-Chat to investigate this dependency for the audio script writing. We prompted Llama2-70B-Chat to generate audio scripts from the storytelling benchmark. The generated audio scripts were fed into the script compiler, which validated their accuracy. We compared the *Compilation Success Rate (CSR)* of Llama2-70B-Chat with that of GPT-4 used in WavJourney. Detailed experimental setup are described in the Appendix A.7.1 in the revision.
>
> In short, Llama2-70B-Chat initially had a low CSR of 10%, which then improved to 54% after the removal of hallucinated content, GPT-4 achieved a high CSR of 94%. These results indicated GPT-4's good performance in audio script writing and suggested that potential enhancements in open-source LLMs may be required to match GPT4’s capabilities in complex tasks such as audio script writing. We also show and discuss several errors made by Llama2-70B-Chat and GPT-4 in the revision.
>
> >”The script writer acts like an agent. There are numerous works and existing methods working on connecting LLM to perform various tasks, e.g., AutoGPT. But the authors didn’t compare against these agent-based system design.”
>
> We appreciate the reviewer's insightful comments. Existing works, such as AutoGPT, focus on using LLMs as agents to perform various tasks, these typically involve using LLMs as deterministic task solvers (e.g., calling APIs, question answering). To the best of our knowledge, the use of LLM agents as creators of engaging, realistic media content remains largely unexplored. WavJourney specifically addresses this gap. WavJourney is designed for compositional audio creation, engaging LLMs in a specific and complex process. As WavJourney asks LLMs to create scripts with structures *(creative)* that serve for Python code generation *(deterministic)*, which makes direct comparisons between WavJourney and AutoGPT challenging.
>
> However, in WavJourney, we propose and design a hand-crafted script compiler to generate Python code from audio scripts. This approach is similar to previous work using LLMs to generate code for complex vision-language deterministic tasks via simple in-context prompting [1, 2]. To further validate our design, we conducted additional comparative experiments between a GPT-4-based script compiler and our proposed hand-crafted script compiler (see Appendix A.7.2 in the revision). The results demonstrate that our approach mitigates instability in script compilation and significantly improves computational efficiency. We believe this experiment provides additional insights into system-level design ablation studies.
>
> Additionally, we acknowledge the importance of comparing our approach with several very recent LLM-based video creation agent designs, such as [3, 4]. We plan to explore this in our future research.
>
> [1] Surís, Dídac, Sachit Menon, and Carl Vondrick. "Vipergpt: Visual inference via python execution for reasoning." *arXiv preprint arXiv:2303.08128* (2023).
>
> [2] Gupta, Tanmay, and Aniruddha Kembhavi. "Visual programming: Compositional visual reasoning without training." *Proceedings of the IEEE/CVF Conference on Computer Vision and Pattern Recognition*. 2023.
>
> [3] Lin, Han, et al. "Videodirectorgpt: Consistent multi-scene video generation via llm-guided planning." *arXiv preprint arXiv:2309.15091* (2023).
>
> [4] Lian, Long, et al. "LLM-grounded Video Diffusion Models." *arXiv preprint arXiv:2309.17444* (2023).

---

### Official Review · Reviewer_HBpU · 2023-11-02

**Soundness:** 2 fair
**Presentation:** 3 good
**Contribution:** 2 fair
**Rating:** 5
**Confidence:** 3

**Summary:**

- The paper presents WavJourney, a system for compositional generation of audio assets by leveraging existing specialized audio models and combining them with LLM-based agents.
- WavJourney is composed of an LLM-based audio script writer which generates an audio script given a text instruction from a user. This script is organized in a JSON formatted object which captures the spatio-temporal relationships between different audio ‘events’ which would be one of speech, music, or sound effects.
- A script compiler takes the audio script and generates a computer program that is executed to call one of AudioGen, MusicGen, or Bark models. There is a mechanism to generate background and foreground audio in the script.
- The authors also propose an Audio Storytelling benchmark using ChatGPT and come up with various evaluation criteria such as engaging, creativity, relevance, emotional resonance, and pace & tempo. Listeners rate audio generated by a system from 1-5 for each of these criteria.
- The authors evaluate WavJourney on text to audio generation and the aforementioned storytelling benchmark. They use objective and subjective metrics for the text to audio evaluation. The results show that WavJourney performs the best on the text-to-audio generation, and received scores marginally about moderate ratings on the storytelling benchmark. The authors also emphasize that the results of WavJourney are better than ground truth data of the AudioCaps dataset.

**Strengths:**

- The paper proposes an interesting approach for human-machine co-creation specific to audio. To the best of my knowledge, there is no other system that is leveraging existing LLMs and audio generation models for such a use-case. The use of a deterministic compiler for generate the exact instructions from the output of the LLM is an interesting choice. Additionally, the choice to focus separately on foreground and background seems to work quite well in enhancing the quality of the outputs.
- While I am not very impressed with the results especially in terms of naturalness, I can see that such an application might be useful for storytelling.

**Weaknesses:**

- There doesn’t seem to be substantial technical contribution in the paper. The main technical components I can see are:
  1. Using the in-context learning capabilities of LLMs to generate well-formatted outputs for audio scripts.
  2. Converting the scripts to calls to 3 different models.
  3. Carefully assembling the generated audio into a single unit.

  While I do appreciate the effort it takes to assemble all these components, I am not too sure of their scientific value to the ICLR community.
- Based on the text, it is not clear what the basis of the metrics proposed in Audio Storytelling benchmark is. There are no references to any studies from the relevant communities which justify the choice of the specific criteria. I would expect the authors to at least add some information, either in the paper, or an appendix, going into more detail regarding the justification being choosing these metrics.
- The result showing WavJourney outputs to be better than AudioCaps GT audio seems a little off to me.
  1. It’s not clear what the preference test is focusing on: the overall quality, or the adherence to the caption.
  2. I wonder if the authors ran the GT audio through the same audio quality enhancement model for evaluation.
  3. In table 3, the 6th column seems to have the wrong entry bolded. GT audio has a higher REL score than WavJourney audio.
  4. The fact that the results are so one-sided for Clotho dataset gives the impression that the result is not too signification (even though the statement is true), or that there is something wrong in one of the two datasets.

  Based on the samples shared, I felt all the GT audio to be more natural sounding than WavJourney. I would encourage the authors to also showcase examples where WavJourney was rated higher than GT by the listeners.
- The generated results have pretty severe audio artifacts. This is not necessarily a weakness of WavJourney itself, but is an issue with the audio generation models themselves.

**Questions:**

- See weaknesses section.
- I would also be interested in seeing a Turing-test type question asking users if they can tell whether the audio is generated or not.

---

> ### Author Response · Authors · 2023-11-23
> **Response to Reviewer HBpU**
>
> We sincerely thank you for your insightful feedback! Please see the following responses to the questions.
>
> >”what the basis of the metrics proposed in Audio Storytelling benchmark is”
>
> The proposed metrics is inspired by expert-driven metrics design process [1]. Specifically, we develop these metrics with input from audio and product experts. These five metrics (creativity, relevance, emotional resonance, engaging, pace & tempo) allow for an assessment of audio storytelling that goes beyond traditional evaluations (e.g., Mean Opinion Score) to consider narrative complexity, music and sound design, and vocal delivery in a coherent and comprehensive manner. We have added these texts in the Section 4 in the revision.
>
> [1] Shah, J.J., Smith, S.M. and Vargas-Hernandez, N., 2003. Metrics for measuring ideation effectiveness. Design studies, 24(2), pp.111-134.
>
> >”It’s not clear what the preference test is focusing on: the overall quality, or the adherence to the caption.”
>
> The preference test is designed to evaluate the overall impression of the generated audio. This evaluation is supported by evidence from A/B testing [2, 3] a widely accepted method for comparing two versions of a variable to determine which performs better. In our case, the A/B tests consistently showed that listeners preferred audio that not only had high-quality sound but also closely matched the provided caption.
>
> [2] Siroker, D., & Koomen, P. (2013). A/B testing: the most powerful way to turn clicks into customers. John Wiley & Sons.
>
> [3] A/B Testing: https://en.wikipedia.org/wiki/A/B_testing
>
> >”I wonder if the authors ran the GT audio through the same audio quality enhancement model for evaluation.”
>
> We didn’t run VoiceFixer for GT audios as it will suppress all non-speech sounds.
>
> >”In table 3, the 6th column seems to have the wrong entry bolded. GT audio has a higher REL score than WavJourney audio.”
>
> Thanks for your suggestion! We have fixed it in the revision.
>
> >“The fact that the results are so one-sided for Clotho dataset gives the impression that the result is not too signification (even though the statement is true), or that there is something wrong in one of the two datasets.”
>
> We would like to highlight that the improvements achieved by WavJourney over AudioGen are considerable (WavJourney uses AudioGen), as evidenced by both subjective and objective results on the Clotho dataset, and subjective results on the AudioCaps dataset, including the additional subjective tests (ablation studies, Turing test) we conducted, as reported in the Appendix A.5 and A.6 in the revision.
>
> WavJourney achieved negative objective results on AudioCaps dataset compared with AudioGen. However, we argue that objective evaluation metrics, such as FAD, may not always effectively capture the performance of text-to-audio generation systems. This viewpoint aligns with conclusions drawn in previous work [4], which suggests that systems with good subjective evaluation results may not always achieve high FAD scores.
>
> [4] Choi, Keunwoo, et al. "Foley sound synthesis at the dcase 2023 challenge." *arXiv preprint arXiv:2304.12521* (2023).
>
> >”I would also be interested in seeing a Turing-test type question asking users if they can tell whether the audio is generated or not.”
>
> Thanks for your constructive suggestions! It’s interesting to see if WavJourney can generate audio that is indistinguishable from real-world audio in human perception, considering its good results achieved on text-to-audio generation benchmarks. To do so, we conduct both Turing test on AudioCaps and Clotho dataset. To mitigate any potential bias that might skew listeners’ perceptions, we provide human participants with audio clips only (without texts). The experimental setup and results are detailed in the Appendix A.6 in the revision. The conclusion we made is, WavJourney outperforms AudioGen and AudioLDM systems in Turing test score, but can not pass the Turing test, indicating that there is a gap between WavJourney generated synthetic and real audio as judged by human listeners. The discrepancy between the results of the Turing test and other subjective metrics (e.g., OVL, REL, preference-based test) on AudioCaps may be attributed to text-to-speech generated speech. Although this generated speech can positively influence subjective metrics, it may be easily to distinguish due to human's sensitive perception of speech.
>
> >”Based on the samples shared, I felt all the GT audio to be more natural sounding than WavJourney. I would encourage the authors to also showcase examples where WavJourney was rated higher than GT by the listeners.”
>
> Thanks for your suggestions! We have uploaded more results to the project page (https://anonymous.4open.science/w/WavJourney_Anonymous/).

---

### Official Review · Reviewer_koFK · 2023-11-08

**Soundness:** 3 good
**Presentation:** 3 good
**Contribution:** 2 fair
**Rating:** 5
**Confidence:** 4

**Summary:**

The paper presents a framework called WavJourney for compositional audio creation. The WavJounrey first provides high-level text prompts to LLMs to generate an audio script that contains the detailed content description of speech, music, and sound effects. Next, a specific script compiler is used to convert the audio script into computer programs which will execute various pre-trained models to generate and compose the audio content. The framework is training-free and a novel subjective multi-genre storytelling benchmark is proposed to evaluate the generated results.

**Strengths:**

1) The generated samples are well done.
2) The framework is training-free.
3) The framework offers high interpretability and flexible ways to create audio content.
4) A novel subjective evaluation metrics are proposed.

**Weaknesses:**

1) While the generated results are impressive, this work focuses more on production than academic research. It shows how best we can achieve when combining state-of-the-art models, and the contribution is limited from the perspective of technical novelty.
2) Since the SOTA generative models are not perfect everywhere, sometimes those models might not generate the expected content. It seems like the proposed framework doesn't take it into consideration and may not always be reliable or robust to use.
3) As an AI audio creation tool, it would be nice to enable the editing of existing audio content. It happens a lot in movie or TV creation. In this case, the framework needs to take the audio as an input as well. However, it seems like the current framework cannot support that.
4) The evaluations are limited to audio captioning tasks. The evaluation of speech and music qualities is missing.

**Questions:**

1) How well does the Wavjourney perform in music and speech benchmarks?
2) What are some failure cases?
3) What is the exact time cost to produce an audio sample?

---

> ### Author Response · Authors · 2023-11-23
> **Response to Reviewer koFK**
>
> Thank you for the useful feedback and all these valuable questions! Please see the following responses to the concerns & questions below.
>
>
>
> >”The evaluations are limited to audio captioning tasks. The evaluation of speech and music qualities is missing. How well does the WavJourney perform in music and speech benchmarks?”
>
> We would like to clarify our focus. Our target is **general compositional audio creation** task, which differs from traditional audio generation with task-specific conditions (e.g., text-to-speech). We focus on the creation of storytelling audio from text, encompassing speech, music, and sound effects, rather than the generation of individual components such as speech and music. As acknowledged by Reviewer jF76: this task opens up a huge space for AI-powered general audio creation.
>
> For evaluation of text-to-audio generation, we have conducted comprehensive experiments on both the AudioCaps and Clotho datasets. Both the AudioCaps and Clotho datasets consist of all speech, music, and sound effects elements. These experiments include objective evaluations (e.g., FAD, KL, IS) and subjective evaluations (e.g., REL, OVL, preference-based test, Turing-test). Furthermore, we have constructed a storytelling benchmark with diverse genres and conducted comparative experiments with the AudioGen and AudioLDM systems (as updated in the revised Table 4). We believe these extensive evaluations adequately demonstrate the effectiveness of WavJourney in a variety of scenarios. We appreciate your suggestion to consider additional benchmarks in future work.
>
> >”As an AI audio creation tool, it would be nice to enable the editing of existing audio content. It happens a lot in movie or TV creation. In this case, the framework needs to take the audio as an input as well. However, it seems like the current framework cannot support that.”
>
> We appreciate your suggestion to enable the editing of existing audio content, a feature that would indeed be valuable in applications such as movie or TV creation. This work focuses on the compositional audio creation from text, which is a novel and relatively unexplored task to the best of our knowledge. We will study the compositional audio editing task in the future.
>
> >“What are some failure cases”?
>
> Some failures are that sometimes audio generative models (e.g., text-to-audio) produce unexpected results.
>
> >”What is the exact time cost to produce an audio sample?”
>
> To analyze the time cost, we randomly selected 20 text captions from the AudioCaps test set and calculated the average inference time required by WavJourney and AudioGen to generate 10-second audio clips. For WavJourney, we report the time costs associated with script writing and audio generation. The testing was conducted on a machine equipped with a single NVIDIA GeForce RTX 2080 Ti GPU. The results are shown in Table below and added to the revision. In practice, by carefully optimizing the use of computational resources (e.g., parallel inference) for WavJourney, it is possible to achieve high-quality synthetic audio generation while minimizing the time required for inference, which we leave as future work.
>
> | Model | Script Writing Time (s) | Audio Generation Time (s) |
> | --- | --- | --- |
> | AudioGen | - | 23.0 |
> | WavJourney | 8.1 | 45.3 |

---

### Official Review · Reviewer_xGho · 2023-11-09

**Soundness:** 2 fair
**Presentation:** 3 good
**Contribution:** 2 fair
**Rating:** 3
**Confidence:** 3

**Summary:**

The authors leverage large language models to connect various audio models for audio creation, such as storytelling audio content with diverse audio elements from textual descritpions.  The separation capabilities of this paper is related to HuggingGPT, combing multiple audio generation models with the user interaction to generate the final audio content. The novelty of this paper is to use LLM's text parsing ability to encode the long user inputs into the format which is easy to process using script compilers, to utlize the audio generation models.

**Strengths:**

The problem is interesting.

**Weaknesses:**

1. The way this paper deals with audio temporal relationships is to generate audio using multiple models and manually connect the generated audio. However, considering the ambiguity of language and the complexity of natural audio, there may be partly overlap among the generated foreground audio lists. This method cannot well fit the distribution of natural audio data. The audio shown in the demo and the mel-spectrogram shown in the article show this shortcoming: there is a clear separation between different audio contents, and there are obvious traces of artificial processing.The author also pointed out this point in the limitations.

2. The author points out that WavJourney outperforms previous state of the art methods in both subjective and objective evaluations. However, on the objective indicators of the Audiocaps dataset, WavJourney's performance is lower than AudioGen's.

3. The experimental part of the article is rather inadequate, The analysis of composite audio generation capabilities of the method is limited to the scene of storytelling, and the analysis of interactive audio creation in multi-round dialogues is also oversimplified, making an inadequate demonstration of the core capabilities of the model.

**Questions:**

1. More ablation study need to be conducted.  Considering that WavJourney uses AudioGen for text-to-audio synthesis, the phenomenon of superiority over AudioGen on objective metrics of the clotho dataset requires further analysis.  Is this due to the introduction of a text-to-speech synthesis model by WavJourney, or is it due to WavJourney's use of LLM to improve its understanding of long captions?

2. More examples of speech content control should be provided. Whether users can control the speech content in the story? It’s important if you want to involve text-to-speech function. I didn't see any relevant examples of speech content control.

---

> ### Author Response · Authors · 2023-11-23
> **Response to Reviewer xGho**
>
> We appreciate the reviewer's constructive and clear feedback! Please see the following responses to the concerns & questions raised in the review.
>
> >“The author points out that WavJourney outperforms previous state of the art methods in both subjective and objective evaluations. However, on the objective indicators of the AudioCaps dataset, WavJourney's performance is lower than AudioGen's.”
>
> As we described in Section 5.2, objective evaluation metrics (e.g., FAD) may not always be an effective way to evaluate the performance of text-to-audio generation systems. This is consistent with conclusions drawn from previous work, which suggests that some systems with good subjective evaluation results may have poor FAD scores [1]. We believe that the subjective evaluation results detailed in Section 5.2 (OVL, REL, and Preference-based test), as well as the additional results provided in Appendix A.5 and A.6 of the revision (ablation studies and Turing tests), are sufficient to demonstrate that WavJourney outperforms AudioGen on the AudioCaps dataset. Additionally, more examples can be found on the project page at https://anonymous.4open.science/w/WavJourney_Anonymous/.
>
> [1] Choi, Keunwoo, et al. "Foley sound synthesis at the dcase 2023 challenge." arXiv preprint arXiv:2304.12521 (2023).
>
> >”Considering that WavJourney uses AudioGen for text-to-audio synthesis, the phenomenon of superiority over AudioGen on objective metrics of the clotho dataset requires further analysis. Is this due to the introduction of a text-to-speech synthesis model by WavJourney, or is it due to WavJourney's use of LLM to improve its understanding of long captions?”
>
> We would like to thank the reviewer for pointing out the importance of conducting ablation studies to explain the superiority of WavJourney over AudioGen. We have conducted ablation studies on **both the AudioCaps and Clotho datasets**. We manually categorized the test audio clips into two splits: 1) Speech-Inclusive, which includes speech and sounds, and 2) Non-Speech Exclusive, which consists only of non-speech elements. The results from the Non-Speech Exclusive evaluation can tell us if the improvements come from the LLM, while the results from the Speech-Inclusive category can show the combined effectiveness of the LLM and TTS models.
>
> Detailed results and analysis are provided in Appendix A.5 of the revision. In conclusion, **WavJourney consistently outperforms AudioGen in OVL and REL metrics for both Speech-Inclusive and Non-Speech Exclusive categories on the AudioCaps and Clotho datasets**. More significant enhancements were observed in the Speech-Inclusive category. These results suggest that WavJourney's advancements are derived from **both the LLM and TTS models**.
>
> >“The experimental part of the article is rather inadequate, the analysis of composite audio generation capabilities of the method is limited to the scene of storytelling”
>
> We would like to clarify that both the AudioCaps and Clotho datasets consist of all speech, music, and sound effects elements. Comparative experiments with state-of-the-art end-to-end generative models such as AudioGen and AudioLDM (as well as ground truth audios) on text-to-audio generation tasks also provide valuable insights into the advancement of our proposed compositional approach.
> We have also included additional results (e.g., Turing tests on text-to-audio and system-level ablation studies) in revision. We hope that these additional results and analysis will address your concerns regarding the adequacy of our experiments.
>
> >“More examples of speech content control should be provided. Whether users can control the speech content in the story? It’s important if you want to involve text-to-speech function”
>
> We have provided an additional study on a radio-play-based storytelling case focusing on voice content control, as shown in Appendix A.9 in revision. In the example, we can control the content of the speech by "topic" or "specific sentences" and the type of language in the multi-turn conversation.
>
> >The analysis of interactive audio creation in multi-round dialogues is also oversimplified, making an inadequate demonstration of the core capabilities of the model.”
>
> We agree that interactive audio creation in multi-round dialogues is important. However, the focus of this work is more on the **compositional generation of audio with sound effects, speech, and music using LLMs**. While existing end-to-end methods often failed in providing a solution for this task - which is important and useful in real-world audio production applications. In future work, we will explore the interactive capability of WavJourney further.

---

### Official Review · Reviewer_iM1V · 2023-11-09

**Soundness:** 2 fair
**Presentation:** 2 fair
**Contribution:** 3 good
**Rating:** 6
**Confidence:** 3

**Summary:**

This paper presented a compositional audio content creation method using LLM and speech, sound effect, and music generation models. The procedure is like this. If the user types specific scenario text, then LLM analyze the text and generate audio script based on the detailed audio prompt templates. This audio prompt templates define the type of the output file format (which is multiple json formats depending on the number of sound sources,) and audio_type (music, speech, or sound effect), duration, volume, foreground/background, position in time, and description of the audio characteristics (e.g. types of music, speech text, or description of sound effect). Then, using this json each for sound source, audio generation models are called and generate each sound source. Finally, the generated sound sources are mix down together. The proposed method is quite novel and has some distinguished characteristics from the most relevant previously proposed method, which is AudioGPT in my opinion. In the previous work, the concept of utilizing audio generation models with chatGPT has been already proposed. However, the previous work mostly focused on calling each generation model separately, so tried to focus on conversational audio generation (single task generation, not the mix of the multiple tasks). Also, in this paper, the author utilized position in time, duration, and volume like characteristics, which further opens up the usage of the LLM on audio model.

**Strengths:**

The paper focused on a compositional audio content creation, not the end-to-end generation model. In audio domain, I believe that this kind of compositional audio content creation will be more useful, because audio has specific physically proven time-frequency relationship, and is really weak for noise (in human perception), so that, a compositional creation can be one solution. And, the paper described one way to do that by tackling in sound types, sounds level, sounds position, etc.

**Weaknesses:**

I think this paper is more like a positional paper rather than an experimental paper. The authors evaluated the proposed methods in two ways. One for validating the proposed method, they evaluated the proposed method on already established text2audio generation task. This evaluation showed that the proposed compositional audio generation method is working well within an audio generation task which I think is enough to show the effectiveness of the proposed method. As a second experiment, the authors further explored audio storytelling creation task. And, in my opinion, this experiment is more like what we can do more or further use cases rather than a solid evaluation method. They measured five subjective metrics, however, since the generated audio content contains music, effect, and speech sounds, we are not sure about where the impressions are came from. Also, baseline is not existed (even though it's not for this task, comparing with audiogen and audioldm might give some baselines).

**Questions:**

Therefore, I think if the authors can give more insights, details about experimental setup, and analysis on the proposed methods rather than describing audio storytelling evaluation, would give a lot more insights to the readers. For example, as the authors mentioned in Section 3.2, there are quite not small possibilities that LLM doesn't follow the instructions. The authors mentioned that the proposed script compiler reduced the instability of LLMs. If the authors can explain this kind of part in more detail, by adding the average possibility of the error case with/without script compiler, and how the script compiler is built, then the readers can get more insights about the practical use of the method.

---

> ### Author Response · Authors · 2023-11-23
> **Response to Reviewer iM1V**
>
> We thank the reviewer for the constructive and clear feedback, which will undoubtedly enhance our work. In response to the feedback, we have included new results and analysis & insights in the revision:
>  - A detailed description of the script compiler implementation can be found in Appendix A.4.
> - We have conducted experiments and comparative analysis of GP4-based script compiler and proposed hand-crafted script compiler, as shown in Appendix A.7.2. The results demonstrate that our approach effectively mitigates instability in script compilation and greatly improves computational efficiency.
> - We have included quantitative experiments, analysis, and a study of error cases on the use of open-source LLMs such as Llama-70B, as described in Appendix A.7.1.
> - Ablation studies and Turing tests on text-to-audio generation can be found in Appendices A.5 and A.6, respectively.
> - We have also further explored interactive audio creation in multi-round dialogues, as shown in Appendix A.9.
>
> Regarding the storytelling creation task:
> - Our motivation for introducing this task is to address real-world audio production scenarios. In practice, people often begin with an abstract or simple concept rather than a detailed idea (description) for each component (e.g., speech, music, audio). This is a common yet challenging scenario that existing audio generative models struggle to handle effectively. With recent advances in LLMs, we now have the potential to automate this task. Our aim is to establish the task with appropriate evaluation metrics and benchmark baseline performance for future research.
> - We compared WavJourney with AudioLDM and AudioGen systems and found that WavJourney outperforms them in all subjective metrics. To ensure consistency during subjective evaluation, we discarded the results in the submission version and re-evaluated the results. The updated experimental results can be found in Table 4 of the revision. We have also uploaded the generated audios of these baselines to the project page for listening (https://anonymous.4open.science/w/WavJourney_Anonymous/).
> - We appreciate your concern about the source of the impressions in our evaluation. Our proposed five metrics, inspired by the expert-driven metrics design process outlined in Shah et al. [1], are designed to capture the holistic impression of the generated audio content, which includes music, sound effects, and speech. Specifically, our subjective evaluation protocol is developed with input from audio and product experts, considering narrative complexity, music and sound design, and vocal delivery in a coherent and comprehensive manner. Therefore, the impression captured by each metric are derived from the overall audio experience, rather than any single component. We have updated the descriptions in the Section 4 in the revision.
>
> [1] Shah, J.J., Smith, S.M. and Vargas-Hernandez, N., 2003. Metrics for measuring ideation effectiveness. Design studies, 24(2), pp.111-134.

---

### Author Response · Authors · 2023-11-23
**Response to concerns regarding academic/scientific value**

Dear Reviewers,

Thank you for your thoughtful and detailed feedback on WavJourney. We value your concerns regarding its academic contribution and would like to address them from four aspects, highlighting the scientific value our research brings to the academic community:

1. **Audio Creation**: WavJourney introduces a novel approach to audio creation that goes beyond domain-specific conditions. By leveraging LLMs with multiple expert audio models, WavJourney can generate audio with diverse elements such as speech, music, and sound effects from textual descriptions. To the best of our knowledge, WavJourney is the first LLM-based agent for audio creation. Our work greatly broadens the scope of existing AI-powered audio generation.
2. **Text-to-Audio Generation**: Compared to published state-of-the-art works on text-to-audio generation [1, 2, 3], we have conducted more extensive subjective evaluations on two benchmarks, including OVL, REL, Preference-based tests, and Turing tests, with sufficient ablation studies and analysis (as updated in the Appendix). These evaluations demonstrate the effectiveness of our compositional approach for text-to-audio generation. We believe WavJourney can provide valuable insights for researchers actively working on this emerging task.
3. **Storytelling Benchmark**: In real-world audio production applications, people often begin with abstract or simple concepts rather than detailed ideas for each component (e.g., speech, music, audio). This is a common yet challenging scenario that existing audio generative models struggle to handle effectively. With recent advances in LLMs, we now have the potential to automate this task. This is our motivation to introduce the new task. Our work established an audio storytelling task with appropriate evaluation metrics and provided baseline performance. We believe our work is insightful for the audio community and hope we can inspire further research in this direction.
4. **Compositional Approach for Multimodal Content Creation**: Our work presents a compositional approach for content creation that is not limited to audio but can be potentially extended to other modalities. The future of content creation is undoubtedly multimodal (including video, text, audio, speech, music, etc.), and using an end-to-end model presents many challenges and may not easily produce acceptable results. We are excited to see very recent works on this approach for video creation [4, 5]. We believe WavJourney provides valuable insights into future research of agent-based multimodal content creation.

We are grateful for the opportunity to clarify these aspects of our work and thank you again for your constructive feedback. We are committed to continually enhancing the impact of our research in the scientific community.

[1] Kreuk, Felix, et al. "Audiogen: Textually guided audio generation." (ICLR 2022).

[2] Liu, Haohe, et al. "Audioldm: Text-to-audio generation with latent diffusion models." (ICML 2023).

[3] Yang, Dongchao, et al. "Diffsound: Discrete diffusion model for text-to-sound generation." *IEEE/ACM Transactions on Audio, Speech, and Language Processing* (2023).

[4] Lin, Han, et al. "Videodirectorgpt: Consistent multi-scene video generation via llm-guided planning." *arXiv preprint arXiv:2309.15091* (2023).

[5] Lian, Long, et al. "LLM-grounded Video Diffusion Models." *arXiv preprint arXiv:2309.17444* (2023).

---

### Meta-Review · Area_Chair_jCJF · 2023-12-11

**Metareview:**

This paper was reviewed by three experts and received mixed scores. Though all reviewers agree some aspects of the paper are promising, they also consistently raise concerns listed below.

1. The technical novelty/contribution of this work is limited (iM1V,  HBpU,  jF76).

2. The experiments are limited. More baseline comparisons and examples are required to verify the effectiveness of this model (xGho, koFK).

While the research demonstrated indeed has promise, the decision is not to recommend acceptance in its current state. The authors are encouraged to consider the reviewers' comments when revising the paper for submission elsewhere

**Justification For Why Not Higher Score:**

1. The technical novelty/contribution of this work is limited (iM1V,  HBpU,  jF76).

2. The experiments are limited. More baseline comparisons and examples are required to verify the effectiveness of this model (xGho, koFK).

**Justification For Why Not Lower Score:**

NA

---

### Decision · Program_Chairs · 2024-01-16

Reject